# Functional aspects of the Eustachian tube by means of 3D-modeling

**Robert Schuon** [1]*, **Josef Schwarzensteiner**[1], **Gerrit Paasche**[1,2], **Thomas Lenarz**[1,2], **Samuel John**[3]

**1** Department of Otorhinolaryngology, Hannover Medical School, Hannover, Germany, **2** Hearing4all Cluster of Excellence, Hannover Medical School, Hannover, Germany, **3** HörSys GmbH, Hannover, Germany

* schuon.robert@mh-hannover.de

## Abstract

The extent of dysfunction of the Eustachian tube (ET) is relevant in understanding the pathogenesis of secondary otological diseases such as acute or chronic otitis media. The underlying mechanism of ET dysfunction remains poorly understood except for an apparent genesis such as a nasopharyngeal tumor or cleft palate. To better describe the ET, its functional anatomy, and the biomechanical valve mechanism and subsequent development of diagnostic and interventional tools, a three-dimensional model based on thin-layer histology was created from an ET in this study. Blackface sheep was chosen as a donor. The 3-D model was generated by the coherent alignment of the sections. It was then compared with the cone-beam computed tomography dataset of the complete embedded specimen taken before slicing. The model shows the topographic relation of the individual components, such as the bone and cartilage, the muscles and connective tissue, as well as the lining epithelium with the lumen. It indicates a limited spiraling rotation of the cartilaginous tube over its length and relevant positional relationships of the tensor and levator veli palatine muscles.

**Data Availability Statement:** Repository: Functional Aspects of the Eustachian Tube by Means of 3D-Modeling https://doi.org/10.5281/zenodo.5102632 Files (1,7 GB) contain the raw

## Introduction

Inadequate function of the Eustachian tube (ET) causes middle ear ventilation disorders. In an epidemiologically relevant disease with 2 million patients in the US every year, further research into its pathogenesis is important [1]. The ET is a biomechanical valve between the nasopharynx and the middle ear. Physiologically, it controls the passive adaptation of the middle ear air pressure to the ambient air pressure primarily via direct muscular actions of the soft palate. In the closed state it protects the middle ear [2]. Form and function are mutually dependent, and this is evident in functional anatomy and biomechanics.

In some cases, pathogenesis is distinct. For example, adenoid vegetations and other benign lesions [3] but also nasopharyngeal carcinoma and other malignant entities can additionally increase the ET opening resistance by local pressure on the cartilaginous part of the ET [4] or displacement of the entrance, by structural ingrowth with destruction, or neuromuscular impairment [3]. Further, a cleft palate formation [4–7] as a result of an embryonic malformation in the palate area usually leads to a disturbed muscular action, such that the ET cannot

data, primary image data including segmentations, mutually spatially reconstructed image data, and quantitative evaluations. Data are available under the terms of the Creative Commons Zero "No rights reserved" data waiver (CC0 1.0 Public domain dedication).

**Funding:** This study was supported by BMBF RESPONSE – partnership for innovation in implant technology, FKZ 03ZZ0902E (awarded to TL) and HörSys GmbH (SJ). The funders provided support in the form of salaries for authors [SJ], but did not have any additional role in the study design, data collection and analysis, decision to publish, or preparation of the manuscript. The specific roles of these authors are articulated in the 'author contributions' section. HörSys GmbH is also a beneficiary of the joint-research grant "BMBF RESPONSE – partnership for innovation in implant technology, FKZ: 03ZZ0928C." No money was transferred from HörSys GmbH to the authors of the other institutions. HörSys GmbH only paid the salary of its own employees coauthoring this paper. HörSys GmbH has no product or service that is part of this study and publication.

**Competing interests:** I have read the journal's policy and the authors of this manuscript have the following competing interests: SJ is employed by HörSys GmbH. This does not alter our adherence to PLOS ONE policies on sharing data and materials. All other authors have declared that no competing interests exist.

open when yawning or swallowing to compensate for pressure differences between the tympanum and ambient air pressure. Other obvious reasons for tubal dysfunction can be impaired nasal breathing due to enlarged nasal conchae, septal deviation or chronic sinusitis [8–10], which can be detected endoscopically or by imaging. In addition, there is evidence for correlations of decreased epitympanal middle ear ventilation of a narrower formation of the tympanic isthmus [11]. But in many cases, the clinician has not such apparent local findings.

Due to its eminent clinical relevance, the ET has been investigated in various other studies for descriptive issues. Macro- [12–15] and microscopic anatomical studies [16] were carried out describing the compartments involved down to the cellular level. Endoscopic studies enable the processing of functional questions in vivo, partly coupled with electromyographic, acoustic, or pneumatic measurements [17–21]. Imaging procedures without dissecting procedures increasingly provide insight into the anatomy and physiology of the tube [22–24]. Tomographic multiplanar imaging is especially the subject of static and dynamic functional studies both from bony parts of the ear [25] and of the ET [11, 25–29]. In addition, optical coherence tomography might present new possibilities for three dimensional imaging [30]. Also, three-dimensional modeling [25, 31, 32] and simulations such as finite element methods [7, 33, 34] have been applied. 3D modeling now makes it possible to investigate functional aspects in very small anatomical functional units, such as when examining the saccus endolymphaticus in Meniere's disease patients [25]. Furthermore, other endoluminal diagnostics [35, 36] and therapies of the ET such as balloon dilatation [37–39] and stenting [40, 41] of the ET are becoming increasingly important in clinic and research. The studies complement and support each other in different ways. Although a large number of studies on the ET are available, a detailed and comprehensive functional description based on the anatomic structures is not yet available.

A detailed three-dimensional microanatomy study was conducted in the present study to better understand these functional aspects. The part controlling the function of the ET is the cartilaginous part, which mainly consists of a lining of respiratory mucous membrane of the lumen, supporting cartilage, Ostmann fat pad (OFP), musculature as well as connective tissue attachment sites and positional relationships between each other. The muscular interaction between the fixed points of the hard palate with the pterygoid process and the posterior attachment to the base of the skull via the soft palate is essential, and is supplemented and supported by actions of jaw movement and tongue-pharynx activity. The soft palate as a posterior continuation of the hard palate is formed in a mirror-symmetrical construction [42, 43], an upper muscle ring towards the base of the skull and a lower muscle ring towards the tongue and pharynx sidewall. The essential muscles of the soft palate involved in ET opening are the tensor veli palatini muscle (TVPM) and the levator veli palatini muscle (LVPM), which act on elements of the ET via divergent force vectors. Locally adjacent muscles, such as the tensor tympani muscle with close positional relationship parallel to the bony part of the ET, are insignificant for tube function [44].

To functionally decode the complex arrangement and, if necessary, to handle it for simulation purposes, two essential prerequisites are required: (1) high-resolution imaging to differentiate the individual compartments clearly and (2) a consistent and quantitatively evaluable image data set in three dimensions. To date, transmitted light microscopy of thin sections is the gold standard for histological examination in pathology. Here, quantitative analyses are already established by digitizing sectional images [41]. Generating a three-dimensional volume model from two-dimensional sections (in the sense of the slices) requires a relational assignment of the individual sections. This proper allocation requires data of the correct spacing on the different sections. Also, it is essential to stack the digitized image sections according to the axis to prevent systematic errors due to torsion [45]. Blackface sheep were chosen as an animal

model for this study due to the relative similarity of their ET to the human ET [46]. A quantitatively evaluable three-dimensional model of the ET and its functional elements in qualitative spatial conformation might be an essential element in understanding the valve mechanism and the subsequent development of new therapies.

## Material and methods

### Ethics approval

The State Office for Consumer Protection and Food Safety, Dept. of Animal Welfare approved experiments with blackface sheep including the use of ET for histological evaluations following German and European animal welfare legislation under the numbers 13/1089 and 13/1283.

### Sample collection and preparation

An unscathed, untreated right ET of a fresh carcass of a blackface sheep (no. 211) from the above-mentioned study 13/1283 was dissected. Preparation of the tissue followed the protocol provided by Pohl et al. [36]. Briefly, after dissection, the specimen was fixed in formalin (3.5%, pH 7.4; C. Roth, Karlsruhe, Germany) for two weeks. In contrast to the earlier published protocol [41], before embedding, three Sterican® standard needles (0.9 mm x 70 mm; B. Braun Melsungen AG, Melsungen, Germany) were placed by hand in the soft tissue as parallel as possible to the course of the tube as landmarks for the later reconstruction of the sample. Care was taken to remain with the needles outside the cartilaginous ET. The tissue block was dehydrated by using an increasing ethanol series (70%, 80%, 90%, 100%; Merck KGaA, Darmstadt, Germany) and embedded in methyl methacrylate (MMA; Merck). The received block was additionally referenced for further control with the milling of three opposing grooves.

### Series cutting, staining, and digitization

Due to the height of the block with the embedded ET, it had to be divided into two blocks of half the height for cutting with the hole saw (Leica—SP1600®, Leica Biosystems Nussloch GmbH, Nussloch, Germany). Cone-beam computed tomography (CBCT) was performed with both halves (XORAN xCAT®, Xoran technologies, MI, USA, ENT scan, high-resolution 0.3 mm). Subsequently, 100 sections of 33 μm thickness were produced with the hole saw at equal distances of 330 μm (the thickness of the sawing blade). After each section, the remaining thickness of the respective block was measured at the three grooves. Two sections were lost due to mounting of the blocks on the sample holder (Fig 1a). Sections were stained with methylene blue (Loeffler's Methylene blue solution; Merck) for 45 s at 80 ˚C and alizarin red (Alizarin red S staining solution; Merck) for 1.5 min. Digitization of the sections was performed using a digital microscope (Biorevo BZ-9000®, KEYENCE, Osaka, Japan) at 2x magnification and a resolution of 3094 x 4094 pixels.

### Data preprocessing and image stacking

For each section, electronic white balancing was performed using the open source software ImageJ (release 1.49m). For the data preprocessing, the positions of the three cannulas, placed approximately parallel to the tube axis, were marked in each section for fiducial registration. Additionally, compartments of histological structures were segmented such that musculature, cartilage, mucosa, bone, and OFP were individually defined in each section.

To generate a 3D-model of the ET, all stained and segmented sections were serially merged into a three-dimensional dataset in stereo lithography, or surface tessellation/triangulation language-format (STL-format) and arranged based on the thickness of the individual parallel

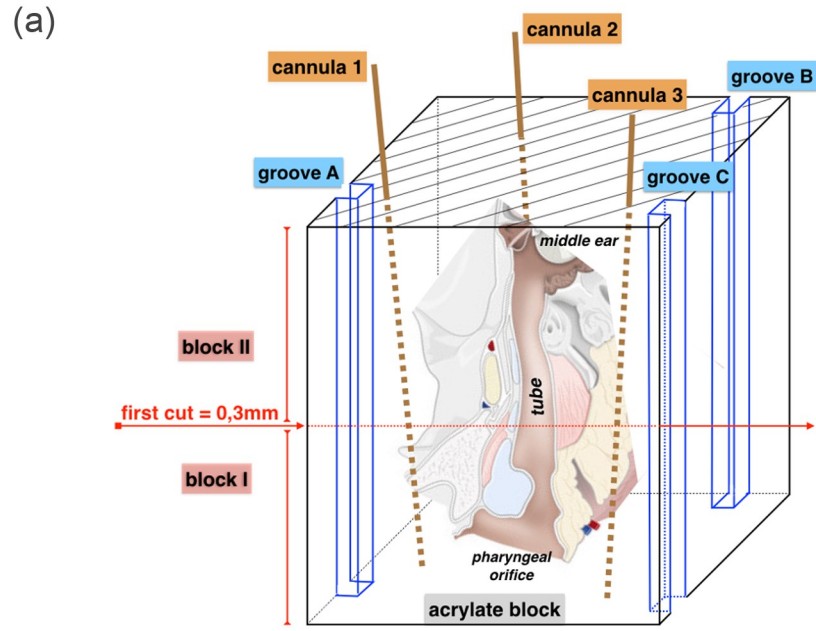

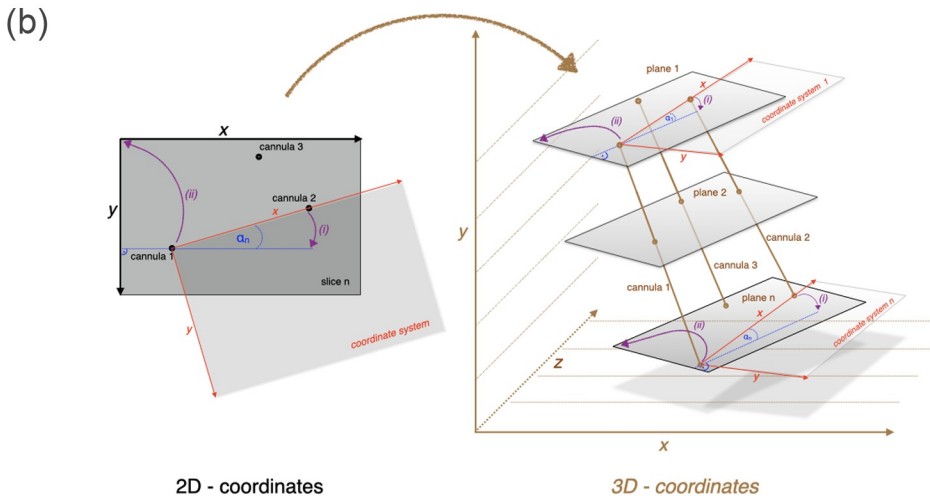

2D - coordinates                    3D - coordinates

**Fig 1. Generation of the 3D model.** From the embedded block to a coherent stack of sections. The block with the embedded tube was cut into two halves. Grooves A to C and cannulas 1 to 3 were used to facilitate orientation after sectioning. The positions of the cannulas are indicated by dotted lines in the volume of the acrylate block (a). Decisive for the creation of a valid volume data set is the axially correct alignment of the digitized sections one after the other: adjustment and stacking of the different sections according to position of the cannulas (b).

sections (33 μm) and the thickness of the saw blade (330 μm) as spacing between the individual sections (z-axis). The open source software platform 3DSlicer® (https://www.slicer.org/, release 4.4) was used to work with this dataset. The orientation of the first section of each of the two blocks was corrected due to the mounting in the hole saw. Next, the axial alignment of the histological images was carried out utilizing the position of the three cannulas (x, y-axis) (Fig 1b) in each image. The stacked sections were then registered with the CBCT scans. The positions of the cannulas and the grooves were used as landmarks for the stacking and for the registration. To achieve best fits between both datasets and least deviations from linearity of

cannulas and grooves, individual sections had to be slightly tilted to account for not perfect parallelism of the sections stemming from the sawing process.

## Results

The segmented sections could be stacked into a consistent volume data set (Fig 2) that allowed for multiplanar imaging and quantitative analysis. The thin section preparation and staining (compare Fig 2B) allowed for proper differentiation and segmentation of the individual compartments in the different sections. Due to this segmentation, the different compartments could be extracted and visualized from the 3D model of the ET (Fig 3). Sometimes compartments appeared partly perforated. After checking the situation in adjacent sections these were, where appropriate, combined to form a single functional unit. The registration of the three-dimensional histological data set with the CBCT DICOM data set shows good agreement throughout the whole stack (Fig 4). The close relationship between the cartilaginous ET and the bony base of the skull can be seen together with the course in the sulcus tubarius as well as the entry into the bony ET (Fig 4B). The positional alignment of the thin slice sections shows relational regularities, such as the spatial configuration of the cartilaginous parts of the ET itself and the positional relationship to the crucial muscles, TVPM, and LVPM. The quantitative analysis of the tube cartilage, lumen and OFP is provided in Fig 5. The cross-sectional area of the tube cartilage remains largely constant over the area under consideration. However, the circumference of the tube cartilage decreases abruptly before entering the isthmus region of the ET and posterior to the area where TVPM and LVPM are connected to the cartilage, bony or connective tissue. The measured lumen of the ET is in most parts small with a large circumference indicating a more or less closed tube. Only at the pharyngeal orifice there was some opening detected. Close to the isthmus, the area of the lumen increased, and the circumference decreased indicating an open lumen at the isthmus and further laterally. Area and circumference of the OFP decreased more or less continuously towards the isthmus.

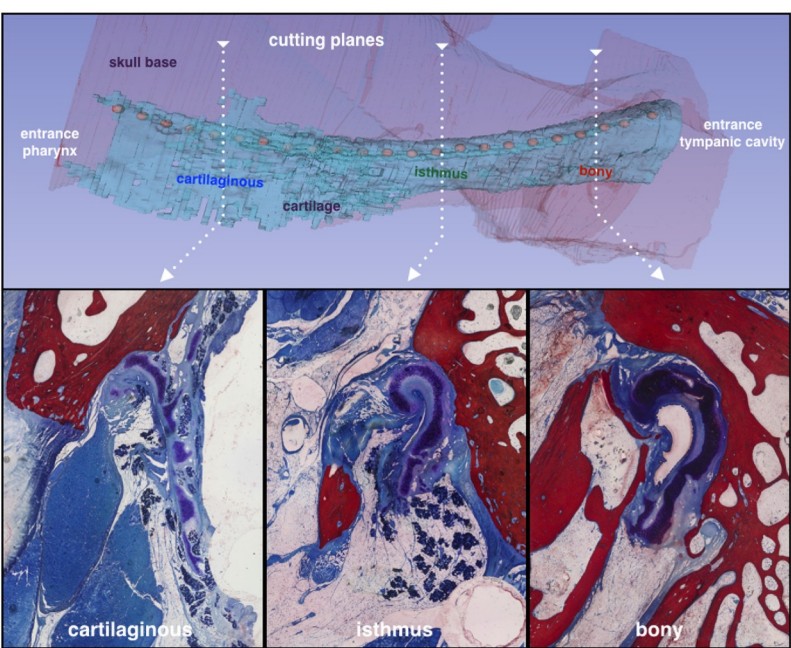

**Fig 2. Three-dimensional model.** The different sections of the ET are combined to form a 3D representation of the ET (top). Bottom: Examples of individual sections from the different parts of the ET.

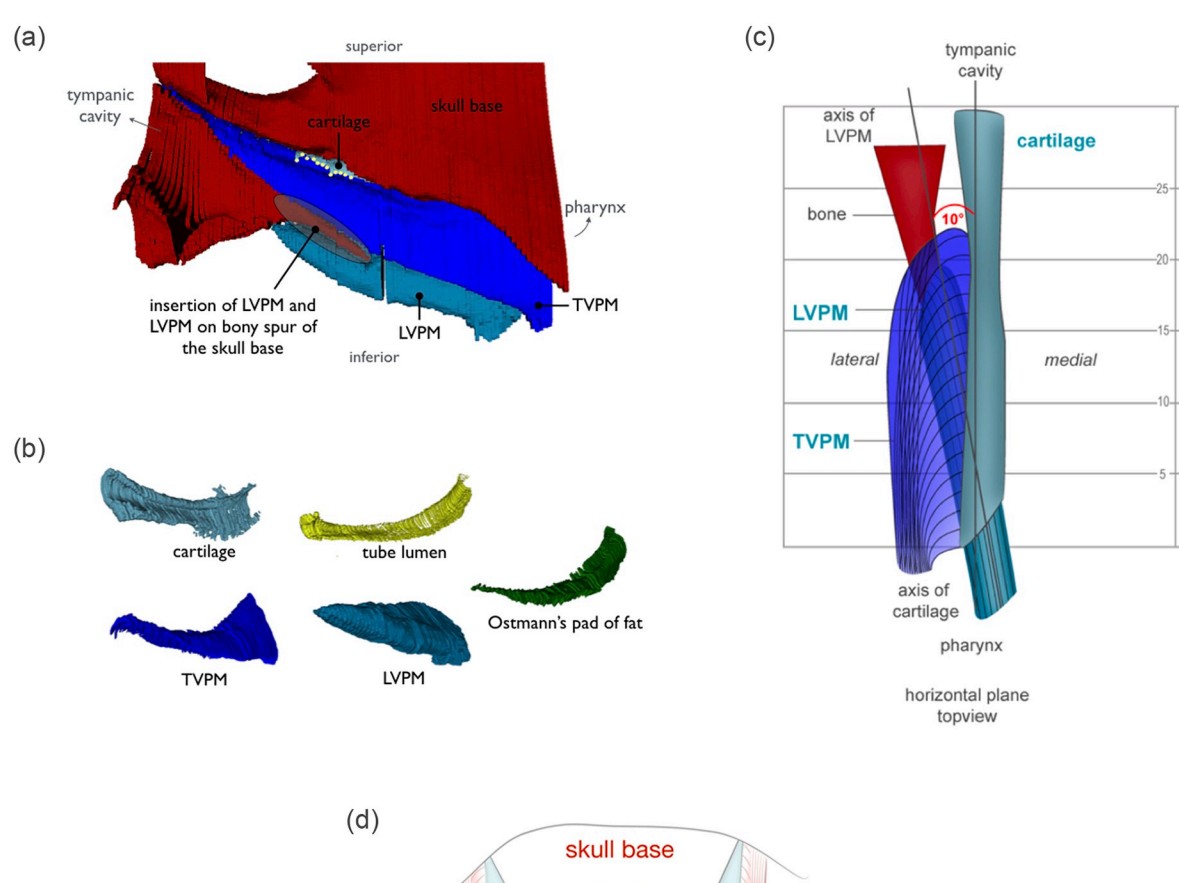

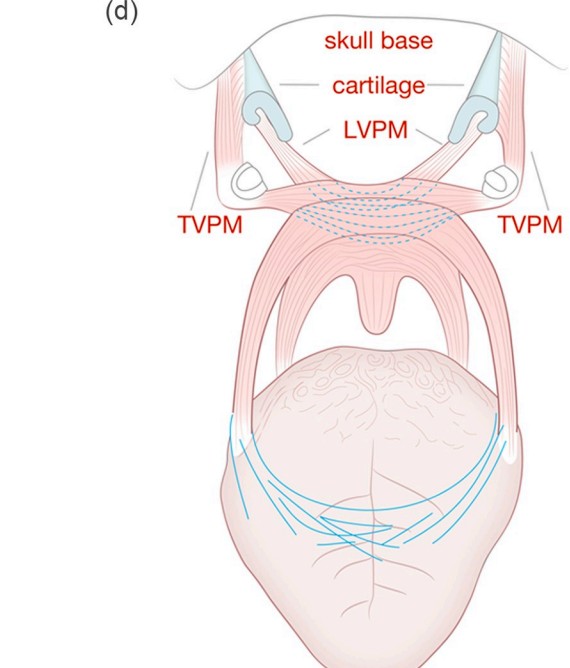

**Fig 3. Compound model of the ET from lateral side (a) and isolated compartments (b) in correlation to the bony skull base (red).**
From left: cartilage (petrol), ET lumen (yellow), TVPM (dark blue), LVPM (light blue), OFP (green). (c) Compound model of the ET above, skull base removed. Bony process of the skull base as origin of the LVPM in red and other compartments in same colour as Fig 3. Compare to coronal projection in Fig 7 illustrating spiral rotation of tubular cartilage. (d) Topographic schematic of the major muscular functional components involved in tube opening. The loop of the LVPM forms a functional muscular circuit. The TVPM inserts in the lateral part on both sides of the tubal cartilage.

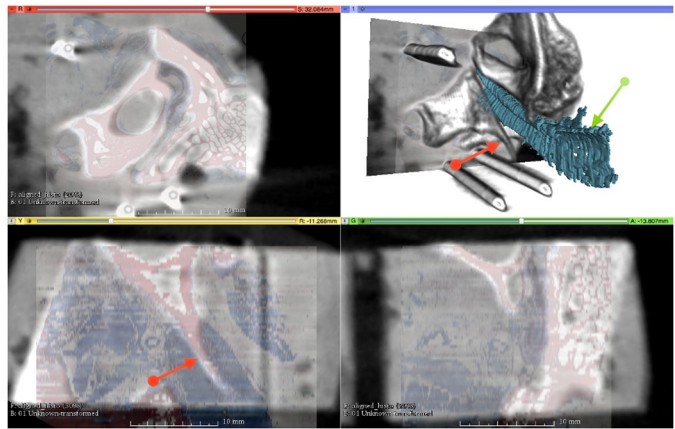

**Fig 4. Fusion of the histologic 3D model and the CBCT.** Note the marking cannulas for reference in the reconstruction (top right) with superimposed tubal cartilage (petrol). The three multiplanar para-planes with superimposed histological section (colored) into the CBCT (grey) come from the upper left (coronary, corresponding histologic section—plane of segmentation: red: bone, blue: soft tissue) over lower left and lower right (sagittal and axial, with reconstructed histological section: slice thickness in the form of jumps). Red arrow: corresponding bony process of the skull base. Green arrow: cartilage of the ET.

The LVPM undercuts the lower edge of the medial tubal cartilage (Fig 6). The course of the LVPM in relation to the inferior margin of the ET cartilage is scissor-like from posterior to anterior is directed towards the middle. In cross-view the muscular shape of LVPM is wedge-like. The TVPM inserts in direction to the middle ear in the lateral arm of the ET cartilage, as well as the skull base. The tube cartilage, when oriented on the long medial arm, shows a torsion from the middle ear to the pharynx of 38˚ (Fig 7).

## Discussion

A three-dimensional model of ET of blackface sheep was created using the digitization of large-format histological sections and their spatially correct stacking. Geometrically, the model corresponds to the CBCT produced before sectioning. The confirmed quantitative evaluation is therefore permissible. The segmentation of the functional entities, such as muscles, cartilage, bone, and connective tissue, allows the three-dimensional visualization as well as measurement of the different compartments and their relationships. The connection of the tubal cartilage with the bony skull base with its course in the sulcus tubarius, and the transition into the bony section of the ET can be shown topographically (compare Fig 4B). In other anatomical studies, a quantitative analysis of entities has also been performed but the relational assignment was based on landmarks such as the characteristic paisley-shape of the cartilage in cross-sections [12]. This results in the image of a more or less uniform canal in a three-dimensional projection. Our model demonstrates rotation of the tubular cartilage, which we believe is essential for the biomechanical opening function in the interaction of the muscles involved (Fig 7). Early studies describe the hourglass-shaped configuration of the ET with the extension to the pharyngeal orifice of the lumen [47]. This can be confirmed by the results of the current study in sheep and can also be supplemented by the specification of the shape of the tube cartilage. A decreasing circumference of the cartilage, with a cross-section that remains uniform at the same time, shows an increasingly compact shape of the cartilage in the direction to the isthmus and the bony part of the ET. The more compact shape of the cartilage towards the middle of the ET suggests an increased bending stiffness compared to the thinner cartilage formation in the direction of the pharyngeal orifice. The structural mechanism of the ET seems functionally

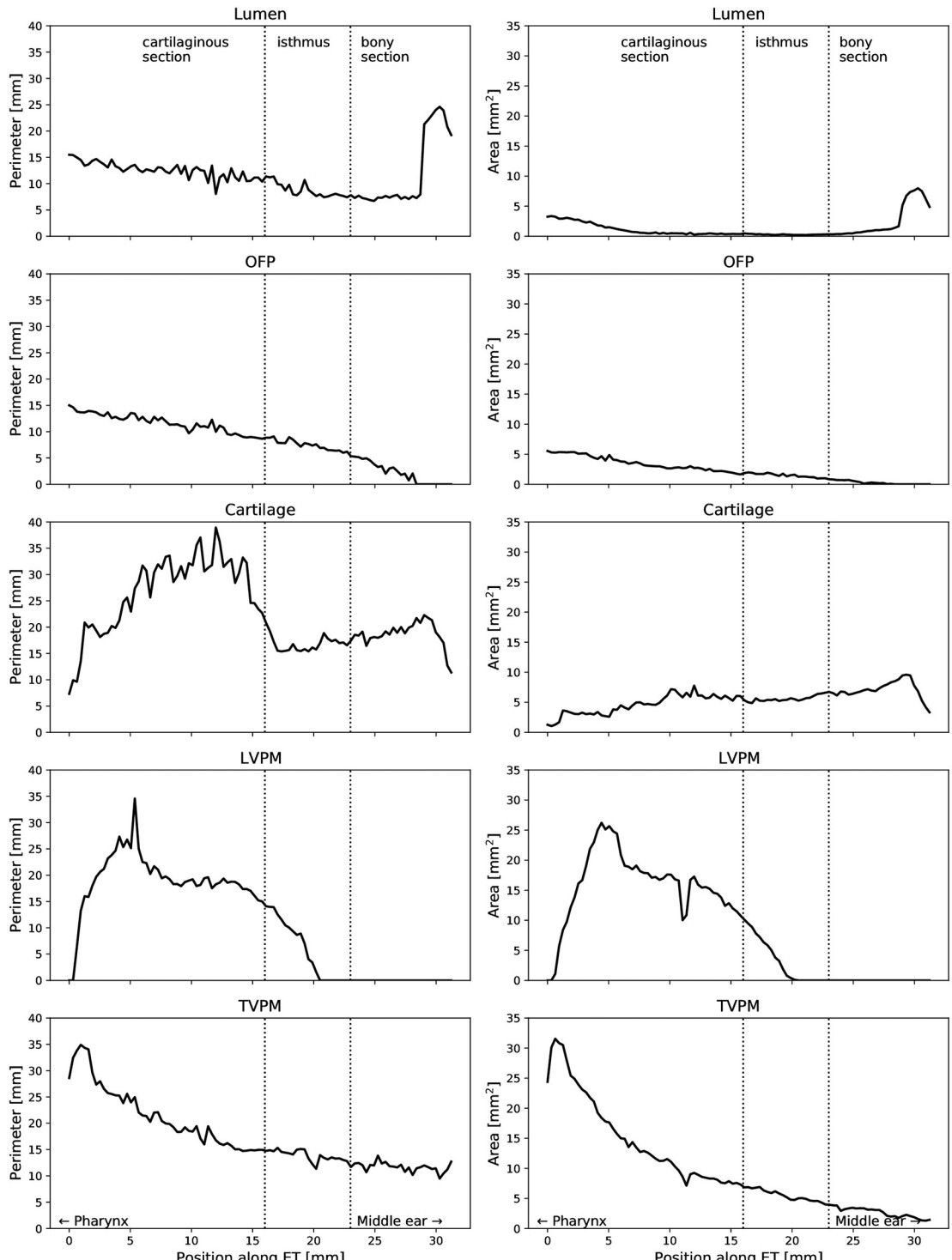

**Fig 5. The plots show the quantitative analysis as a function of the position in the ET axis with respect to area and circumference for the individual entities cartilage, lumen, OFP, TVPM and LVPM (a).**

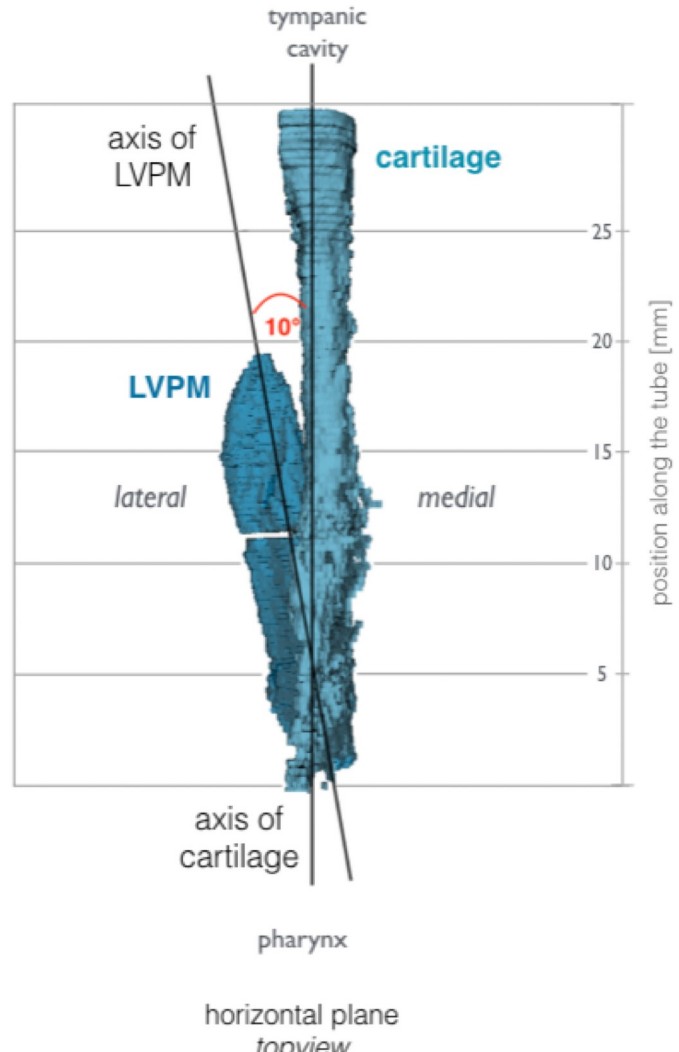

**Fig 6. View from above on a right ET, pharyngeal orifice below.** Relation of the tubal cartilage and the course of the LVPM. By the elevation of the soft palate by yawning and swallowing the muscle slides in the cartilage, initially from the pharyngeal opening to the upper third of the cartilage. Together with the upper-outward rotation with additional opening momentum by the force of the TVPM, the opening of the ET is initiated.

similar to the statics of hollow cylinders cantilevered perpendicular to the ground to withstand lateral forces, comparably with modern skyscrapers.

The small open lumen and large circumference indicates that the cartilaginous tube is largely closed coming from the pharyngeal orifice. In the region where the cartilage becomes more compact, the circumference of the lumen also gets smaller but without affecting the area as seen in cross sections. Only when approaching the isthmus, the area of the lumen increases again. As this goes along with the smallest circumference it is taken as indicator for a small but permanently open lumen. The axial diagram of the course corresponds to the examination of the ET at rest in the OCT [30]. The large circumference is caused by the longitudinal mucosal folds [48] along the tube axis, which are predominantly posterior to the torus tubarius up to the transition area [14].

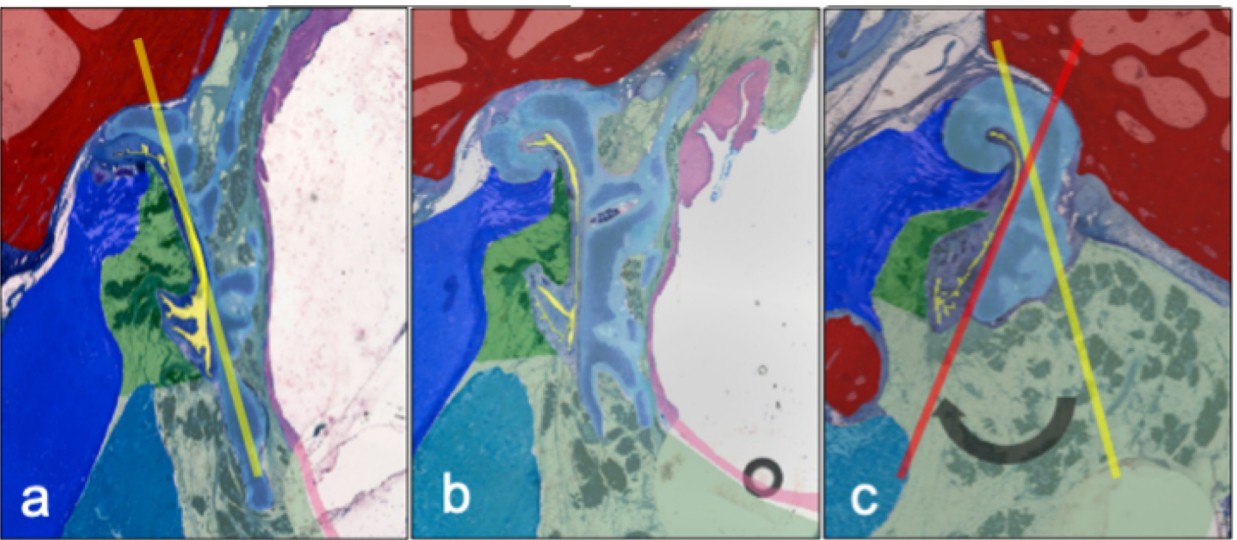

**Fig 7. ET-cartilage shows a spiraled conformation in the direction of its axis.** The caudal edge of the cartilage turns medially in the direction from nasopharynx (a) over midportion (b) to middle ear (c) (angle of approximately 38˚). Note the circle in b) corresponding to the needle in this section. Also shown is the bone (red), the TVM (dark blue) LVPM (light blue) with its origin of the bony process of the skull base.

A detailed macroscopical morphological study of the muscular entity, such as the TVPM, inevitably requires the removal of other relevant structures involved in dissection, such as the LVPM [13]. From a surgical point of view, this is certainly valuable, but this can only contribute to a limited biomechanical understanding. The difficulty of microscopic morphological studies using parallel histological section series which allow axial image rotation by referencing individual landmarks, such as the typical paisley-shaped cartilage axis in ET, can be a methodical source of error. Different positions of the LVPM to the medial tube cartilage may have arisen because the rotated position of the respective incision in the ET course led to a different position of the LVPM [49]. In this study, it was shown that the cartilage of the ET in the direction from the bony portion to the soft palate not only forms a uniform groove-like structure from the back craniolaterally to the front caudomedially, but is also, in this direction, rotated slightly helical inwards around its own axis. Functionally, this is relevant because the LVPM is located in the lifting portion of the soft palate [21] with its wedge-shaped profile caudal lateral of the lateral cartilage and thus not only raises this but also performs a rotation of the lower end towards the sagittal plane. The wedge-shaped muscle part in the cross-section of the LVPM is located near the pharyngeal orifice. In the area of the bony origin in the skull base near the isthmus, the muscle belly is rounded. Further studies with human specimen are necessary to check this muscular configuration; in different publications, the muscle belly is always described as rounded in cross-section [14, 49]. The principle of lever forces here might have a significant influence on effectiveness. The larger distance from lumen and LVPM to TVPM has already been described as significant in the adult population compared to children [32]. At the approximately simultaneous activity of the TVPM, which approaches the over the pivot point, the hamulus, proceeding from the caudal and lateral direction directly to the short arm of the ET cartilage, an impulse opposite to the LVPM might arise. A fascial suspension of the ET at the overlying skull base apparently provides for a further axial location stabilization of the ET. This explains the videoendoscopically described mechanism of rotation of the medial lamina and fixation of the lateral lamella [18]. Electromyographic studies of LVPM and TVPM show an initial and longer activation of the LVPM and secondary activation of the TVPM [50].

This might result in the following mechanism. An earlier further inward torsion of the medial cartilaginous lamina by the LVPM is followed by secondary activation of the TVPM which results in a contrary momentum of the short cartilaginous lamina, resulting in a swing open momentum of the cartilaginous groove. Even though the TVPM is commonly referred to as the *dilator tubae*, a relevant proportion of the LVPM might be entirely attributable to the biomechanical hypothesis described here. This was also confirmed by electromyographic analyses [19]. In a study, an opening movement of the ET was assumed, which was triggered by an isometric contraction of the LVPM with a displacement of the medial cartilage [49] of the tube. The position of the LVPM in the cartilaginous tube is essentially dependent on the movement of the muscular double ring, which the ET moves from the hard palate back up towards the base of the skull and back down towards the tongue and throat. This results in different moments of influence on the ET. Typically, the adjustment of the middle ear air pressure takes place passively during swallowing. The latter results in a lifting movement of the soft palate [51]. This in turn leads to a scissor-like swing in of LVPM into the medial cartilage of the tube. The axis of rotation of this movement lies in the bony attachment point of the LVPM laterally and just below the tube. The LVPM undercuts the medial cartilage of the tube. The spirally twisted ET cartilage seems to adapt in a spiral rotation this upward movement of LVPM harmoniously. There is corresponding compliance to the movement of the mucous membrane by primary elasticity and the clear surface surplus in the closed resting position by folding. In addition to the LVPM-mediated opening movement of the medial arm, the j-folded ET cartilage is spread by the TVPM-mediated counter movement of the short arm in a lateral direction.

An even more detailed segmentation of the tissues, for example, the goblet cells [52], could be used in further studies to clarify questions about factors of a possible additional mechanical etiology of chronic ET dysfunction by obstruction.

In this study, the OFP is presented in three dimensions, allowing an analysis of the topographic anatomy embedded in the overall structure. Previous analyses showed a careful evaluation in the axial layer only [16]. This study, which includes a three-dimensional modeling, allows more realistic modeling and also to take a further step towards more detailed analyzes, e.g. finite element methods [16] or mechanical experiments [53]. Based on the findings in the present study, we may speculate that in order to adjust the air pressure in the middle ear to ambient air pressure, the muscular activity in interaction with bony and cartilaginous supporting tissue structures and soft tissue structures, which like the OFP, contribute to the sealed tube, is in principle comparable to a hydraulic valve.

All results presented here were obtained from an animal model. We can refer to the basic structural similarity of the anatomy [43] but for a translation of the results and conclusions to humans, human Eustachian tubes should be evaluated in a comparable manner. An extended investigation with possibly other staining methods for a better representation of fascial structures within the ET could also provide further insights into functional aspects, e.g. principles of tensegrity [54].

## Author Contributions

**Conceptualization:** Robert Schuon.

**Data curation:** Robert Schuon.

**Formal analysis:** Samuel John.

**Funding acquisition:** Gerrit Paasche.

**Investigation:** Robert Schuon, Josef Schwarzensteiner.

**Methodology:** Robert Schuon, Samuel John.

**Project administration:** Robert Schuon, Gerrit Paasche.

**Resources:** Samuel John.

**Software:** Josef Schwarzensteiner, Samuel John.

**Supervision:** Robert Schuon, Gerrit Paasche, Thomas Lenarz, Samuel John.

**Validation:** Robert Schuon.

**Visualization:** Josef Schwarzensteiner.

**Writing – original draft:** Robert Schuon.

**Writing – review & editing:** Gerrit Paasche, Thomas Lenarz.

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
