## [Decision Letter · Decision Letter 0]

22 Feb 2021

PONE-D-20-39610

Functional Aspects of the Eustachian Tube by Means of 3D-Modeling

PLOS ONE

Dear Dr. Schuon,

Thank you for submitting your manuscript to PLOS ONE. The reviewers felt that the manuscript is interesting and merits publication; however, they recommended minor reviews to potentially increase the quality of the study. Therefore, we invite you to submit a revised version of the manuscript that addresses the points raised during the review process.

We look forward to receiving your revised manuscript.

Kind regards,

Rafael da Costa Monsanto, M.D.

Academic Editor

PLOS ONE

Journal Requirements:

2.Thank you for stating the following in the Financial Disclosure section:

"This study was supported by BMBF RESPONSE – partnership for innovation in implant technology, FKZ 03ZZ0902E."  

We note that one or more of the authors are employed by a commercial company: HörSys GmbH

Additional Editor Comments :

Reviewers' comments:

Reviewer's Responses to Questions

**Comments to the Author**

1. Is the manuscript technically sound, and do the data support the conclusions?

Reviewer #1: Yes

Reviewer #2: Yes

Reviewer #3: Yes

2. Has the statistical analysis been performed appropriately and rigorously? 

Reviewer #1: N/A

Reviewer #2: N/A

Reviewer #3: N/A

3. Have the authors made all data underlying the findings in their manuscript fully available?

Reviewer #1: Yes

Reviewer #2: Yes

Reviewer #3: Yes

4. Is the manuscript presented in an intelligible fashion and written in standard English?

Reviewer #1: Yes

Reviewer #2: Yes

Reviewer #3: Yes

5. Review Comments to the Author

Reviewer #1: Thank you very much for the opportunity to review the manuscript titled "Functional Aspects of the Eustachian Tube by Means of 3D-Modeling". I congratulate the authors for this quite interesting study.

Reviewer #2: The presen study makes a contribution to the current litereature providing an analysis on the structure of ET using both histopathological and radiological 3D- reconstruction models. Cons of the study is that only one sample was investigated, and it was not a human sample. Pros of the study are the effort on the methodology, and information provided based on the structure of the ET. In the big picture, it deserves publication. However, in the prevoous literature, there exist a few studies (from the laboratory of Michael Paparella) conducted on human temporal bones, constituting 3d reconstruction models. I thimk these publications deserve citation:

1- Epitympanum volume and tympanic isthmus area in temporal bones with retraction pockets.

Monsanto RD, Pauna HF, Kaya S, Hızlı Ö, Kwon G, Paparella MM, Cureoglu S.

Laryngoscope. 2016 Nov;126(11):E369-E374. doi: 10.1002/lary.25937. Epub 2016 Apr 23.

PMID: 27107158

2- A three-dimensional analysis of the endolymph drainage system in Meniere disease.

Monsanto RD, Pauna HF, Kwon G, Schachern PA, Tsuprun V, Paparella MM, Cureoglu S.

Laryngoscope. 2017 May;127(5):E170-E175. doi: 10.1002/lary.26155. Epub 2016 Jul 21.

PMID: 27440440 Free PMC article.

Reviewer #3: This reviewer congratulates the authors in this meticulous work and great presentation. The points below will enhance the importance of this manuscript.

This reviewer is looking forward to their similar work on human cadaver specimens.

Line 87- TVPM is not parallel to the bony part of the ET. It is almost parallel with the cartilaginous part.

Line 202. Authors refer to a hypothesis here, but it is not clear what this refers to. They discuss the current literature on functional aspects and implications of these anatomical relationships. But they have not clearly identified any novel hypotheses. And obviously, this study is far from testing or proving any of the existing hypotheses. They can just provide suggestive information.

Line 208. After using arrows and labels on Figure 2, this will be more clear. It seems that unlike in humans, the cartilage extends into the bony ET beyond istmus.

Line 256. “analyses”

Line 451.Figue 1. Great demonstration of the methods.

Line 463.Figure 2. It is better to use labels in the bottom images, to identify the bone, cartilage, muscles.

Line 468-Figure 3. Additional combined images (top) from different perspectives (top down, bottom up, lateral to medial (perpendicular to this one) will enhance the reader’s perception.

Line 475. Figure 4. Additional fusion images that include the 2 key muscles will be useful.

Line 490-Fig 6. It would be great to have this figure in two other dimensions, perpendicular to the ET orifice axis, to better appreciate the relationship of the LVPM and the ET cartilage, to better emphasize the “functional” implications of this relationship. Need to be clear also is that this is not a functional model, this statis model though has functional implications due to the presumed changes in the vector and the length. This concept is hard for those readers not familiar with the functional anatomy, so need to expand on this in the text and on the figure.

It is also best to include this ET cartilage and LVPM relationship in multiple images starting from the nasopharyngeal orifice extending deeper, to demonstrate how this relative position shifts, for additional supportive information for the presumed “functional” implications.

Such images will be much more useful than those in Figure 5.

6. PLOS authors have the option to publish the peer review history of their article (what does this mean?). If published, this will include your full peer review and any attached files.

Reviewer #1: No

Reviewer #2: No

Reviewer #3: **Yes: **Cuneyt M. Alper

---

## [Author Response · Author response to Decision Letter 0]

26 May 2021

Dear Managing Editor of Plos One, Dear Reviewers,

our work group has focused on further research about the Eustachian tube. We very much appreciate the comments, the suggestions for improvement as well as the references to supplementary literature. 

We have carefully incorporated all suggestions and additions, supplemented illustrations and added the references mentioned. With this, we now hope to be able to publish the work.

As requested, the review comments can be tracked in the revised version.

Our findings showing new insights and we will be pleased if you want to publish our research.

This study was supported by BMBF RESPONSE – partnership for innovation in implant technology, FKZ 03ZZ0902E (awarded to TL) and HörSys GmbH (SJ). The funders provided support in the form of salaries for authors [SJ], but did not have any additional role in the study design, data collection and analysis, decision to publish, or preparation of the manuscript. The specific roles of these authors are articulated in the ‘author contributions’ section.

HörSys GmbH is also a beneficiary of the joint-research grant "BMBF RESPONSE – partnership for innovation in implant technology, FKZ: 03ZZ0928C.”

No money was transferred from HörSys GmbH to the authors of the other institutions. HörSys GmbH only paid the salary of its own employees coauthoring this paper. HörSys GmbH has no product or service that is part of this study and publication.

I have read the journal's policy and the

authors of this manuscript have the following

competing interests: SJ is employed by HörSys GmbH. This does not alter our adherence to PLOS ONE policies on sharing data and materials.

All other authors have declared that no competing interests exist.

Best regards

Dr. Robert Schuon

---

## [Decision Letter · Decision Letter 1]

24 Jun 2021

Functional Aspects of the Eustachian Tube by Means of 3D-Modeling

PONE-D-20-39610R1

Dear Dr. Schuon,

We’re pleased to inform you that your manuscript has been judged scientifically suitable for publication and will be formally accepted for publication once it meets all outstanding technical requirements.

Kind regards,

Rafael da Costa Monsanto, M.D.

Academic Editor

PLOS ONE

Additional Editor Comments (optional):

Reviewers' comments:

Reviewer's Responses to Questions

**Comments to the Author**

1. If the authors have adequately addressed your comments raised in a previous round of review and you feel that this manuscript is now acceptable for publication, you may indicate that here to bypass the “Comments to the Author” section, enter your conflict of interest statement in the “Confidential to Editor” section, and submit your "Accept" recommendation.

Reviewer #1: All comments have been addressed

Reviewer #2: All comments have been addressed

Reviewer #3: All comments have been addressed

2. Is the manuscript technically sound, and do the data support the conclusions?

Reviewer #1: Yes

Reviewer #2: Yes

Reviewer #3: Yes

3. Has the statistical analysis been performed appropriately and rigorously? 

Reviewer #1: Yes

Reviewer #2: Yes

Reviewer #3: N/A

4. Have the authors made all data underlying the findings in their manuscript fully available?

Reviewer #1: Yes

Reviewer #2: Yes

Reviewer #3: Yes

5. Is the manuscript presented in an intelligible fashion and written in standard English?

Reviewer #1: Yes

Reviewer #2: Yes

Reviewer #3: Yes

6. Review Comments to the Author

Reviewer #1: I congratulate the authors for the revision of their paper. The authors have added a substantial contribution to scientific knowledge about Eustachian tube function.

Reviewer #2: All recommendations were addressed carefully, thus, the revised version of the manuscript can be published.

Reviewer #3: Thanks for the revision.

THIS IS TO THE JOURNAL=

I don't think there should be a minimum character count to state that this reviewer accepts this revision

7. PLOS authors have the option to publish the peer review history of their article (what does this mean?). If published, this will include your full peer review and any attached files.

Reviewer #1: No

Reviewer #2: No

Reviewer #3: **Yes: **Cuneyt M. Alper, MD.

---

## [Editor Report · Acceptance letter]

29 Jul 2021

PONE-D-20-39610R1 

Functional Aspects of the Eustachian Tube by Means of 3D-Modeling 

Dear Dr. Schuon:

I'm pleased to inform you that your manuscript has been deemed suitable for publication in PLOS ONE. Congratulations! Your manuscript is now with our production department. 

Kind regards, 

on behalf of

Dr. Rafael da Costa Monsanto 

Academic Editor

PLOS ONE